# Direct measurement of superdiffusive energy transport in disordered granular chains

Eunho Kim [1,2], Alejandro J. Martínez[3], Sean E. Phenisee[1], P.G. Kevrekidis[4], Mason A. Porter [3,5,6] & Jinkyu Yang[1]

Energy transport properties in heterogeneous materials have attracted scientific interest for more than half of a century, and they continue to offer fundamental and rich questions. One of the outstanding challenges is to extend Anderson theory for uncorrelated and fully disordered lattices in condensed-matter systems to physical settings in which additional effects compete with disorder. Here we present the first systematic experimental study of energy transport and localization properties in simultaneously disordered and nonlinear granular crystals. In line with prior theoretical studies, we observe in our experiments that disorder and nonlinearity—which individually favor energy localization—can effectively cancel each other out, resulting in the destruction of wave localization. We also show that the combined effect of disorder and nonlinearity can enable manipulation of energy transport speed in granular crystals. Specifically, we experimentally demonstrate superdiffusive transport. Furthermore, our numerical computations suggest that subdiffusive transport should be attainable by controlling the strength of the system's external precompression force.

[1] Department of Aeronautics and Astronautics, University of Washington, Seattle, WA 98195-2400, USA. [2] Division of Mechanical System Engineering & Automotive Hi-Technology Research Center, Chonbuk National University, 567 Baekje-daero, Deokjin-gu, Jeonju-si, Jeollabuk-do, 54896, Republic of Korea. [3] Oxford Centre for Industrial and Applied Mathematics, Mathematical Institute, University of Oxford, Oxford, OX2 6GG, UK. [4] Department of Mathematics and Statistics, University of Massachusetts, Amherst, MA 01003-4515, USA. [5] Department of Mathematics, University of California, Los Angeles, CA 90095, USA. [6] CABDyN Complexity Centre, University of Oxford, Oxford, OX1 1HP, UK. Eunho Kim and Alejandro J. Martínez contributed equally to this work. Correspondence and requests for materials should be addressed to M.A.P. (email: mason@math.ucla.edu)

P. W. Anderson's 1958 paper[1] on wave dynamics in disordered systems is one of the landmarks of 20th-century physics, and studies of Anderson localization and related phenomena continue to yield fascinating surprises[2]. Over the past decade, there have been amazing experimental advances, in fields such as ultracold atomic physics and nonlinear optics, toward the direct observation of spatial localization and transport in disordered systems[3–5]. There has been simultaneous progress toward achieving a theoretical understanding of the interplay between disorder and weak nonlinearity[6]. However, there has been much less exploration of disorder in strongly nonlinear systems, and many fundamental questions remain open. Specifically, under what conditions is transport subdiffusive or superdiffusive? More generally, how does strong nonlinearity affect localization? Much of the progress has arisen from studies of models with on-site nonlinearities, such as discrete nonlinear Schrödinger and Klein–Gordon models, where the interplay between disorder and nonlinearity yields subdiffusive transport[6–8]. However, recent progress has hinted at a fundamentally different phenomenology in lattices with inter-site interactions (e.g., well-known settings such as chains of Fermi–Pasta–Ulam–Tsingou (FPUT) type[9,10]). In particular, it has been shown numerically that superdiffusive behavior is possible in these systems[11–13]. An intriguing feature in all of the above situations is that disorder (which is traditionally viewed as leading to localization) and nonlinearity (which can cause localization in the form of phenomena such as discrete breathers[14]) can somehow cancel out each other's tendency toward localization, leading to transport.

In this study, we experimentally and numerically investigate energy transport in one-dimensional disordered granular crystals (i.e., granular chains) in a wide variety of regimes, extending from almost linear to strongly nonlinear ones. Granular crystals composed of spherical particles are a popular vehicle for investigating various nonlinear wave features[15–18]. When in contact, two particles interact with each other nonlinearly via a Hertzian interaction[19]: the force–displacement relation in the contact interaction is governed by a 3/2 power law under compression and zero force under tension. In this class of models, one can tune the effective system nonlinearity very precisely by varying the ratio of excitation amplitude to static precompression[18]. We consider a system excited at a granular chain's boundary to investigate how the mechanical energy injected by the external excitation is transported along the chain under the combined influence of nonlinearity and disorder. In our analysis, we use established diagnostics such as the inverse participation ratio (IPR) and the second moment ($m_2$) of the energy[6]. Earlier works have used such quantities to characterize transport in homogeneous, tapered, and diatomic granular chains[20], as well as in disordered granular chains[11,12]. A very recent study examined the relation between chaos and transport properties in granular chains[21].

## Results

**Experimental setup and data**. We describe our experimental setup in Fig. 1. The granular chain consists of 32 spherical particles (see Supplementary Table 1 and Supplementary Note 1). To introduce disorder in a granular crystal, we use various combinations of aluminum and tungsten-carbide particles, which have starkly different densities and elastic moduli (see the Methods section for details). The right end of the chain is constrained by a steel plate with a hole in the center; through this hole, a spherical impactor released from a ramp hits and excites the first particle of the chain. The left end of the chain is blocked by a large sliding mass, which applies a static precompression $F_0$ to the chain via a linear spring. We measure the dynamics of the chain by recording each particle's velocity as a function of time using a laser Doppler vibrometer (LDV)[22].

We control the strength of the system's nonlinearity by applying three different amounts of precompression: 50, 10, and 0 N, which range from almost linear to strongly nonlinear

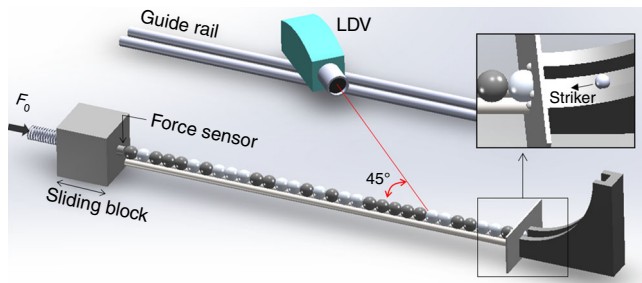

**Fig. 1** Schematic of our experimental setup. The inset shows details of the boundary condition in front of a granular chain

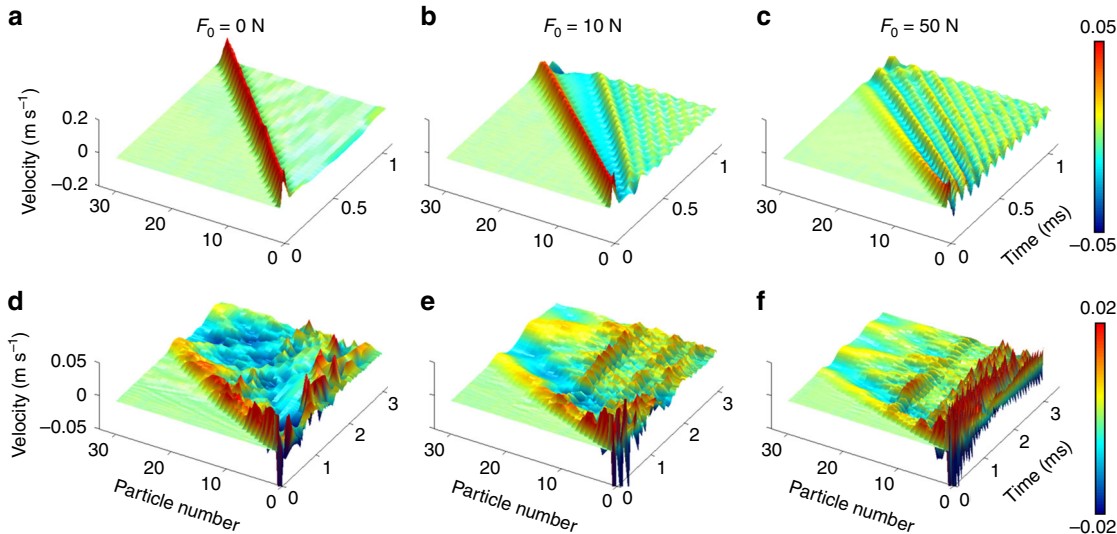

**Fig. 2** Wave propagation in homogeneous and disordered chains. Spatiotemporal distributions of particle velocities in (**a**–**c**) a homogeneous chain and (**d**–**f**) a disordered chain with static precompressions of (**a**, **d**) 0 N, (**b**, **e**) 10 N, and (**c**, **f**) 50 N

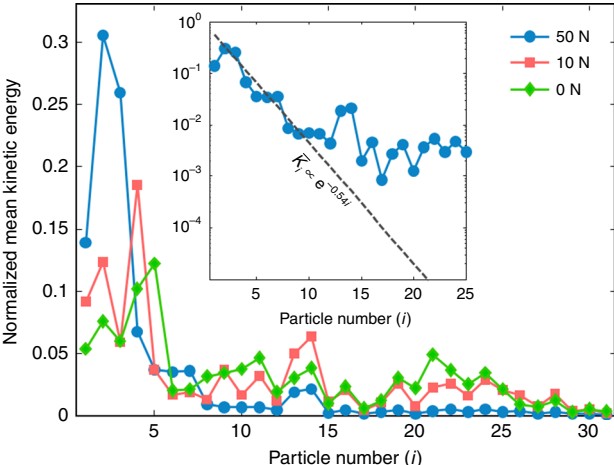

**Fig. 3** Experimental observation of Anderson-like localization. Normalized kinetic-energy profile, averaged between 1.5 and 3.5 ms, for different amounts of precompression. In the inset, we show the kinetic energy on a logarithmic scale when the static precompression is 50 N. The dashed line shows the slope associated with $e^{-0.54i}$, where $i$ denotes particle number

dynamical regimes of the chain. To visualize the propagation of stress waves in each case, we show spatiotemporal distributions of the particles' motions based on the measured velocity. In Fig. 2, we show experimental measurements of particle velocities for a homogeneous chain consisting of (top row) aluminum particles only and (bottom row) a disordered chain composed of aluminum and tungsten-carbide particles in strongly nonlinear, weakly nonlinear, and almost linear scenarios (left, middle, and right, respectively). See Supplementary Fig. 1 and Supplementary Note 2 for details of how we characterize nonlinearity strength.

In a homogeneous chain, we observe a localized wave packet in the form of a strongly nonlinear solitary wave in the absence of precompression (i.e., for $F_0 = 0$ N). When we apply a nonzero precompression to the chain, we start to observe the generation of linear oscillatory waves, which propagate behind the supersonic leading nonlinear wave packet. These ripples arise from oscillations of the first particle after an impact[23] and consequently from the excitation of oscillatory modes. The frequency of these oscillatory waves increases as the precompression increases (compare Figs 2b and 2c). This occurs because the contact stiffness increases as we apply progressively stronger precompression due to the nonlinearity in the contact interactions.

In Figs 2d–f, we show experimental results for wave propagation in a disordered chain for various precompression strengths. (See Supplementary Table 1 and Supplementary Note 1 for details of the disordered chain configuration, Supplementary Fig. 2 and Supplementary Note 1 for our comparison with simulation results, and Supplementary Fig. 3 and Supplementary Note 3 for a corresponding frequency analysis.) Comparing our results for disordered chains with the ones from the homogeneous chain, we find that the presence of disorder causes significant scattering of propagating waves in both time and space. The scattering is most drastic in the absence of precompression (see Fig. 2d). However, for increasing precompression, the wave packet tends to become more localized in the front of the chain and the amplitudes of propagating waves decrease significantly as a function of distance (see Fig. 2f). Given identical excitation conditions, we observe that applying precompression increases the speed of the leading wave packet, as expected in granular chains due to the dependence of the wave speed on wave amplitude and chain stiffness (i.e., precompression)[15–18].

**Experimental observation of Anderson-like localization.** To characterize localization near the excitation point in the almost linear regime (i.e., $F_0 = 50$ N), we compute the kinetic-energy distribution. Given the velocity $v_i(t)$ of the $i^{th}$ particle, we compute its kinetic energy $K_i(t) = (1/2)m_i v_i^2(t)$, where $m_i$ is the particle's mass. We then average our results over the different realizations of disorder to obtain $\langle K_i(t) \rangle$. Initially, the pattern's amplitude decreases due to the spreading associated with non-scattered modes. It then oscillates near the edge of the chain for a long time (up to 3.5 ms). To better visualize the kinetic-energy profile, we compute a temporal mean between $t_s = 1.5$ ms and $t_f = 3.5$ ms. We define the mean kinetic energy as $\overline{K}_i = \int_{t_s}^{t_f} \langle K_i(t) \rangle dt / (t_f - t_s)$, and we then normalize the distribution by letting $\overline{K}_i \to \overline{K}_i / \sum_{j=1}^{N} \overline{K}_j$ (where $N$ is the number of particles). In Fig. 3, we show normalized mean kinetic-energy profiles for the different precompression strengths. In our experiments, we observe for $F_0 = 50$ N that the kinetic-energy distribution seems at first to decay at a roughly exponential rate with particle number: $\overline{K}_i \propto e^{-0.54i}$ for $i \in \{2,...,10\}$. (See the inset of Fig. 3 and Supplementary Note 4 for more details.) Due to this exponential decay, the kinetic energy is reduced by two orders of magnitude after about 10 sites from the boundary of the chain. We also observe localization around the second particle. This particular location arises from a combination of dissipation effects and the particular disordered configurations (from a sub-ensemble in which the first particle is always aluminum) that we examine. For $i \in \{11, ...,15\}$, we observe excitation of a secondary mode that emerges and decays in an irregular manner as a function of time (see Fig. 3). For $i \gtrsim 15$, very low-amplitude waves, which are associated primarily with low-frequency linear modes, reach the right side of the chain and are reflected.

**Energy localization and spreading.** To investigate the energy transport characteristics of the granular chains, we quantify energy localization and the speed of energy spreading using the inverse participation ratio (IPR) and the second moment of the energy ($m_2$), respectively[6,11,24]. To calculate these quantities, we use kinetic energy instead of the total energy of the particles in a chain, because we can directly calculate the former experimentally by measuring particle velocities. The IPR based on the kinetic energy is

$$P^{-1}(t) = \frac{\sum_{i=1}^{N} \left( m_i v_i^2 \right)^2}{\left( \sum_{i=1}^{N} m_i v_i^2 \right)^2}. \qquad (1)$$

When all energy is confined to a single particle, $P^{-1} = 1$, and $P^{-1}$ approaches $1/N$ (where $N$ is the total number of particles in a chain) as a wave disperses. In Fig. 4, we show $P^{-1}$ as a function of time for both the almost linear and nonlinear regimes. Because of the customary exchange between kinetic and potential energies, we observe oscillations in the temporal profile of $P^{-1}$. However, the aggregate trends do not vary significantly from those based on total energy (see Supplementary Fig. 4 and Supplementary Note 5).

In a homogeneous chain with precompression (see the solid blue curve in Fig. 4a), we observe experimentally that $P^{-1}$ decreases in time, because the linear waves disperse as they propagate. However, in a disordered chain (see the solid red curve in Fig. 4a), this decreasing trend is less pronounced. This implies that the disorder tends to favor wave localization, confirming the effect of Anderson localization. Our numerical simulations (dashed curves; see the Methods section) corroborate our

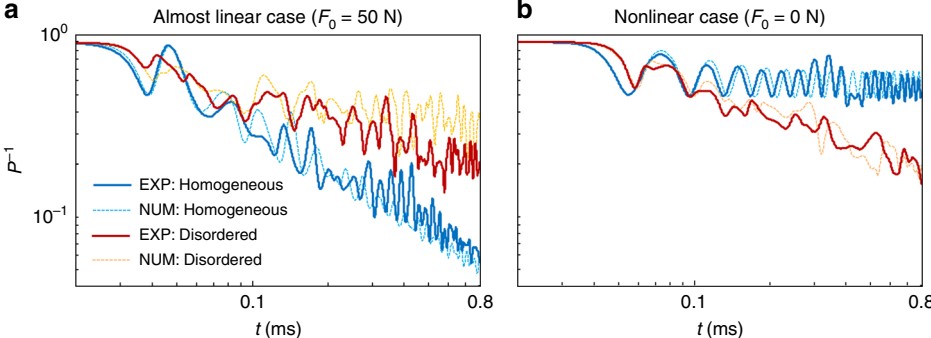

**Fig. 4** Energy-localization characteristics. Inverse participation ratio ($P^{-1}$) of the kinetic energy in homogeneous (blue curves) and disordered (red curves) chains for (**a**) almost linear and (**b**) nonlinear regimes. In both cases, we compare numerical calculations (dashed curves) with experimental data (solid curves)

experimental results. Strikingly, we observe completely different qualitative behavior in the nonlinear regime (see Fig. 4b). The nonlinearity favors wave localization in a homogeneous chain, as indicated by the larger value of $P^{-1}$. In this case, a localized solitary wave propagates in a highly localized (doubly exponentially decaying[25]) manner, and $P^{-1}$ retains a value of about 0.6. A similar trend was noted recently in numerical simulations[11,12] for bulk dynamics in very long chains (so boundary effects were neglected). In contrast, in our experiments, boundary effects are fundamental both for the generation of the initial excitation and for the ensuing dynamics.

The second moment of kinetic-energy distribution is

$$m_2(t) = \frac{\sum_{i=1}^{N} i^2 \left( m_i v_i^2 \right)}{\sum_{i=1}^{N} m_i v_i^2}. \qquad (2)$$

It is well-known that asymptotic energy spreading in a homogeneous chain is ballistic (i.e., $m_2(t) \sim t^2$ as $t \to \infty$)[11,26]. To investigate energy spreading in detail, we estimate the exponent $\gamma$ of the second moment in the scaling relationship $m_2(t) \sim t^\gamma$ during the time period from 0.1 ms to 1 ms[6,11,12]. We compare the estimated exponents from the experimental data to those from numerical simulations in Fig. 5. The exponents in homogeneous chains (see the upper shaded area in Fig. 5) are about $\gamma = 2$ (corresponding to ballistic spreading). For stronger precompression, the exponents are slightly smaller, but they remain near the ballistic regime.

For disordered chains, we observe drastic changes in $\gamma$ as we increase $F_0$. Specifically, the mean values of the exponents for the disordered chains diminish gradually from superdiffusive ones ($1 < \gamma < 2$) to subdiffusive ones ($0 < \gamma < 1$) for chains with progressively stronger static precompression. This is more noticeable in our numerical simulations than in our experimental data; in the latter, we clearly observe the superdiffusive regime, yet our measurements while approaching the diffusive regime do not definitively manifest subdiffusive behavior. In simulations, in contrast, our ability to do a large variety of numerical experiments allows us to collect data for numerous precompression strengths and to transparently reveal the trend from superdiffusion to subdiffusion. It is also important to contrast our subdiffusive numerical results, for which we observe $\gamma \in (0.5, 1)$, with the much slower subdiffusive spreading in discrete nonlinear Schrödinger and Klein–Gordon chains[6–8]. Our experimental results (white squares) indicate that the exponent $\gamma$ of the second moment of the kinetic energy is smaller for progressively stronger

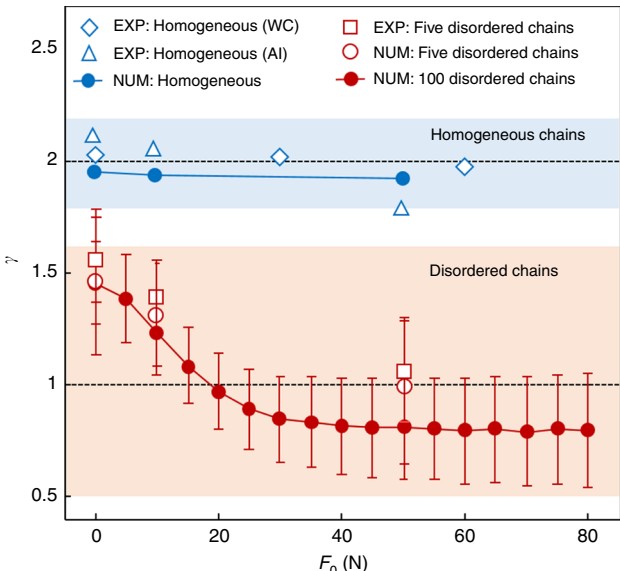

**Fig. 5** Energy-spreading characteristics. Exponents ($\gamma$) of the second moment $m_2$ of kinetic energy as a function of precompression strength. The exponents 1 and 2 (horizontal dashed lines) represent diffusive and ballistic transport, respectively. The diamond, triangular, and square marks are based on experimental data obtained, respectively, from a homogeneous tungsten-carbide (WC) chain, a homogeneous aluminum (Al) chain, and a mean over five disordered chains. (See Supplementary Table 1 and Supplementary Note 1 for details about the five disordered chains.) We mark the numerical data with circles. We do computations for homogeneous chains (blue circles), the five disordered chains (hollow red circles), and 100 randomly generated disordered chains (red circles with error bars). The red circles and error bars, respectively, give the mean values and standard deviations of the computed exponents from the 100 cases

precompression $F_0$, in line with our numerical observations (white circles). However, while our experimental findings definitively manifest superdiffusion, they are inconclusive about the potential transition to subdiffusion as one increases $F_0$.

We also compare the exponents that we obtain from the five disordered chains with those from numerical simulations of 100 different disordered chains in Fig. 5. The red circles and error bars, respectively, give the mean values and standard deviations of the computed exponents from the 100 cases. The relevant trends persist despite the larger error bars for stronger precompressions. The agreement between computations and experiments also continues to hold. Additionally, although our study focuses on

the effect of nonlinearity and disorder, the boundary conditions in granular chains can also alter energy transport mechanisms, as demonstrated previously in other physical systems (e.g., refs. [27,28]). See Supplementary Fig. 5 and Supplementary Note 6 for detailed numerical calculations of such boundary effects.

## Discussion

In this study, we experimentally and numerically investigate energy transport and localization properties in both ordered and disordered one-dimensional granular crystals in strongly non-linear, weakly nonlinear, and almost linear settings. In our experiments, we observe exponential attenuation of the energy distribution for disordered chains when we apply strong pre-compression to obtain almost linear dynamics. This is reminiscent of Anderson localization. We show that this localized pattern oscillates persistently near the chain boundary. For progressively weaker precompression, the system's effective nonlinearity becomes stronger, and there is a gradual progression from a localized pattern to flatter energy distributions. This affects the chain's transport properties, as energy spreads faster for weaker precompression.

Our study paves the way toward using granular crystals and related systems (e.g., ones involving magnets[29]) as accessible platforms for exploring the interplay between nonlinearity and disorder in a lattice setting. The amenability of such granular metamaterials for controllable tuning between strongly nonlinear and almost linear regimes, coupled with the ability to monitor such systems in a distributed fashion, promises a wealth of exciting advances in condensed-matter physics, materials science, and nonlinear dynamics. Proposals to engineer local nonlinearity effects (e.g., see ref. [30]) in granular crystals and to examine the competition between the chiefly superdiffusive effects of inter-site nonlinearities and the apparently chiefly subdiffusive effects of on-site nonlinearities also constitute important future directions.

## Methods

**Experiments.** We measure wave propagation in two types of one-dimensional granular crystals (i.e., granular chains): homogeneous and disordered chains. Each chain consists of 32 particles using either tungsten-carbide (Young modulus $E = 600$ GPa, Poisson ratio $\nu = 0.2$, and density $\rho = 15.6$ g cm$^{-3}$) or aluminum ($E = 69$ GPa, $\nu = 0.33$, and $\rho = 2.8$ g cm$^{-3}$). The radius of each particle is 9.53 mm. The values of $E$, $\nu$, and $\rho$ are based on standard specifications[31]. We examine two homogeneous chains (one made of tungsten-carbide and the other made of aluminum) and five disordered chains (with combinations of tungsten-carbide and aluminum, as described in Supplementary Table 1 and Supplementary Note 1). In the disordered chains, we randomly choose each particle, except for the first particle, as either a tungsten-carbide or aluminum particle, with an independent 50% probability of each for each particle. This yields an 'uncorrelated' type of disorder[11]. As we illustrate in Supplementary Fig. 6 and Supplementary Note 7, a 32-particle granular chain is long enough to validate energy localization and transport properties.

The spherical particles are supported by two stainless-steel rods coated by polytetrafluoroethylene (PTFE) tape to reduce friction between the particles and the supporting rods. We also place two aluminum rods on top of the granular chain, with minimum clearance to restrict lateral motion of the particles. To apply various precompression strengths to a chain, we press a heavy block on a sliding guide toward the chain using a linear spring. Between the block and the chain, we embed a static force sensor (see Fig. 1) to ensure accurate monitoring of the precompression applied to the chains. The other side of a chain is blocked by a steel plate with a circular hole in the center, and we bond four equidistant steel balls to the edge of the hole (see the inset of Fig. 1). These balls have point contacts with the first particle; this yields a much smaller contact damping between the first particle and the plate boundary than one would obtain from a line contact between the two.

For this study, we test granular chains with various precompression strengths (from 0 to 60 N). We give an impact excitation to the first particle using a chrome-steel ($E = 210$ GPa, $\nu = 0.29$, and $\rho = 7.8$ g cm$^{-3}$) particle with a 4.75 mm radius. We roll the particle down a ramp so that it has a normal impact on the first particle through the hole of the plate (see Fig. 1). The impact velocity, which we measure using a laser Doppler vibrometer (Polytec OFV 534), is $0.45 \pm 0.12$ m s$^{-1}$.

To visualize wave propagation in a granular chain, we measure the axial velocities of particles individually for each particle spot using the LDV. To synchronize the measured data, we use a small piezoelectric ceramic plate (3

mm × 4 mm × 0.5 mm) bonded to the first particle; this plate generates a trigger signal (voltage) at the instant of striker impact. We scan the velocity profiles for all particles in the chain by moving the LDV using a moving stage, and we reconstruct the measured data to depict a spatiotemporal profile of propagating waves. For each configuration, we follow the above procedures and conduct three tests. This amounts to 672 experimental realizations to construct the spatiotemporal dynamics of the granular chains for all of the cases that we study in the present article.

**Numerical simulations.** The equation of motion for the $i^{\text{th}}$ particle in a granular chain is[16]

$$m_i \frac{d^2 u_i}{dt^2} = A_{i-1,i} \left[ \delta_{i-1,i} + u_{i-1} - u_i \right]_+^{3/2} - A_{i,i+1} \left[ \delta_{i,i+1} + u_i - u_{i+1} \right]_+^{3/2} - \frac{m_i}{\tau} \frac{du_i}{dt}, \quad (3)$$

where $m_i$ and $u_i$, respectively, are the mass and displacement of the $i^{\text{th}}$ particle. The force on the $i^{\text{th}}$ particle depends on the geometry and material properties of both it and its adjacent particles. The interaction coefficient between the $i^{\text{th}}$ and $(i+1)^{\text{th}}$ particles is

$$A_{i,i+1} = \frac{4 E_i E_{i+1}}{3 \left[ E_{i+1}(1 - \nu_i^2) + E_i(1 - \nu_{i+1}^2) \right]} \sqrt{\frac{R_i R_{i+1}}{R_i + R_{i+1}}}, \quad (4)$$

where $E_i$, $\nu_i$, and $R_i$ are, respectively, the $i^{\text{th}}$ particle's Young modulus, Poisson ratio, and radius. The quantity $\delta_{i,i+1} = (F_0/A_{i,i+1})^{2/3}$ is the compression distance between the $i^{\text{th}}$ and $(i+1)^{\text{th}}$ particles at static equilibrium under a static precompression of $F_0$. The bracket $[x]_+ = \max\{0, x\}$ encodes the fact that there are no tensile forces in interparticle interactions. We determine the damping coefficient $1/\tau$ by measuring the decay of leading waves in experiments and calculating a value for $\tau$ that matches the experimental results. (See Supplementary Fig. 7 and Supplementary Note 8 for further discussion about the effect of dissipation on energy spreading. See Supplementary Note 9 for a brief discussion about testing for plasticity.) For the homogeneous chain of aluminum particles, we use $\tau = 2$ ms (for $F_0 = 0$ N), $\tau = 1$ ms (for $F_0 = 10$ N), and $\tau = 0.4$ ms (for $F_0 = 50$ N). For the disordered chains, we use $\tau = 1$ ms (for $F_0 = 0$ N), $\tau = 0.6$ ms (for $F_0 = 10$ N), and $\tau = 0.4$ ms (for $F_0 = 50$ N).

The equation of motion for the striker particle ($i = 0$) includes an interparticle interaction only with the first particle, and the first particle interacts with the boundary plate and the second particle (in addition to the striker). Similarly, the last particle has an interaction with the penultimate particle and the other boundary plate. The displacements at the boundaries are fixed ($u_{\text{left}} = u_{\text{right}} = 0$), and we apply an initial velocity to the striker ($v_{\text{striker}} = 0.45$ m s$^{-1}$). We conduct numerical simulations using a variety of different impact velocities of about 0.45 m s$^{-1}$ and observe no significant differences in the exponents. More generally, the presence of both superdiffusive transport and wave localization, which we see in the values of both $m_2$ and $P^{-1}$, is robust to changes in the initial condition of the impactor.

We solve the equations of motion (3) numerically with a Runge–Kutta method (of Dormand–Prince type) using Matlab's ODE45 routine with a relative error tolerance of 0.1%. In calculations with lower error tolerances, we obtain the same results. Based on the relative tolerance of 0.1%, the order of magnitude of the error in the displacements is less than $1 \times 10^{-9}$ m, and the order of magnitude of the error in the velocities is less than an amount ranging from $1 \times 10^{-6}$ to $1 \times 10^{-4}$ m s$^{-1}$, depending on the specific properties of the waves.

Using numerical simulations, we probe additional details of disordered granular chains. In addition to those that we discussed earlier, we compare the dynamics in velocity excitations and displacement excitations (see Supplementary Fig. 6 and Supplementary Note 10). We also consider a sub-ensemble in which the first particle is tungsten-carbide. In this case, in contrast to the sub-ensemble in which the first particle is aluminum (for which localization occurs around the second particle), we observe that localization tends to occur around the first particle (see Supplementary Fig. 8 and Supplementary Note 11).

**Data availability.** The data sets generated and analyzed during the current study are available from the corresponding author on reasonable request.

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

## Acknowledgements

A.J.M. acknowledges support from CONICYT (BCH72130485/2013). J.Y. thanks the NSF (CAREER-1553202) for financial support. J.Y. and P.G.K. also acknowledge support from US-ARO (W911NF-15-1-0604) and US-AFOSR (FA9550-17-1-0114), and P.G.K. also gratefully acknowledges support from the Stavros Niarchos Foundation via the Greek Diaspora Fellowship Program. E.K. acknowledges the support from the National Research Foundation of Korea (NRF) grant funded by the Korea government (MSIP, No. 2017R1C1B5018136)

## Author contributions

All authors wrote the paper and contributed equally to the production of the manuscript and interpretation of results; P.G.K., A.J.M., M.A.P., and J.Y. designed the study; E.K. and S.E.P. performed the experiments; and E.K., A.J.M., and S.E.P. did the numerical computations and data analysis.

## Additional information

**Competing interests:** The authors declare no competing financial interests.

