## [Peer Review File · Nature Communications]

Reviewers' comments:

Reviewer #1 (Remarks to the Author):

The authors study energy propagation in granular chains subjected to an impact at one end. The study focuses on disordered chains but includes comparison with the more classical case of homogeneous chains. The system is studied for different degrees of precompression in order to tune the strength on nonlinear effects (nonlinearity being stronger at small precompression). The study combines experimental observations and numerical simulations of a nonlinear mass-spring model based on Hertz contact law. The authors observe superdiffusive or subdiffusive energy transport in the disordered chains (with « uncorrelated » type of disorder), depending whether precompression lies below or above some threshold. The intermediate diffusive regime corresponds to linear growth of the second moment of kinetic energy distribution (in comparison, homogeneous chains lead to a superdiffusive « ballistic » regime corresponding to quadratic growth). This work differs from previous studies on energy transport in nonlinear lattices in several ways : the use of a mixed experimental/theoretical approach, incorporation of dissipative effects in the model (although this is done in a rather simplified manner), and a focus on transient energy transport in finite chains (instead of long-time energy spreading in large systems). This study is very interesting and deserves publication after the following points are addressed :

1-Repeated collision experiments are performed in order to obtain the spatiotemporal distributions of particle velocities. Is it possible that plastic deformations of aluminium beads introduce some bias in the results after too many collision have occurred ?

2-What is the influence of impact velocity on the different power-law behaviors ?

3-At the end of the last section « Methods », it is indicated that the relative error tolerance for ODE45 is set to 10^{-6} m for displacements. Since relative displacements typically lie below one micron in the present physical setting, it may be worthwhile to check numerical computations with a lower error tolerance.

4-It could be interesting to mention the following reference dealing with impacts in dissipative granular chains : N.S. Nguyen and B. Brogliato, Multiple impacts in dissipative granular chains, Lect. Notes in Applied and Computational Mechanics 72, Springer, 2014. In this work (see section 5.3), the inverse participation ratio P^{-1} is computed (in the form of an equivalent coefficient $C_{\{KE\}} = \sqrt{N \cdot P^{-1} - 1}$) for different types of granular chains (homogeneous, tapered, diatomic) after impact termination.

Reviewer #2 (Remarks to the Author):

This is an interesting study of wave propagation in disordered granular chains. It probes Anderson localization, and the destruction of Anderson localization due to nonlinearity, whose strength is controlled with the precompression strength. Anderson localization itself is quite a sensitive phenomenon relying on phase coherence, and it is quite surprising that the authors can demonstrate its existence (Fig.3). The crossover from superdiffusive to subdiffusive spreading with increasing precompression is also nice, although the experimental data are actually not (yet) indicating subdiffusion, and the argument of the authors - similar data for few numerical runs, but systematic lowering of the exponent for more averaging over more runs - is not a solid one, reducing that claim to wishful thinking. Therefore the authors will have to change the title accordingly.

I must also say that fitting power laws over one decade is not very solid. Probably the large error bars

are a consequence of that.

To conclude, this paper can be published, but only after the authors change the title, abstract, and the main body of the paper, removing the claim to have measured subdiffusive transport.

Reviewers' comments:

Reviewer #1 (Remarks to the Author):

The authors study energy propagation in granular chains subjected to an impact at one end. The study focuses on disordered chains but includes comparison with the more classical case of homogeneous chains. The system is studied for different degrees of precompression in order to tune the strength on nonlinear effects (nonlinearity being stronger at small precompression). The study combines experimental observations and numerical simulations of a nonlinear mass-spring model based on Hertz contact law. The authors observe superdiffusive or subdiffusive energy transport in the disordered chains (with α uncorrelated β type of disorder), depending whether precompression lies below or above some threshold. The intermediate diffusive regime corresponds to linear growth of the second moment of kinetic energy distribution (in comparison, homogeneous chains lead to a superdiffusive α ballistic β regime corresponding to quadratic growth). This work differs from previous studies on energy transport in nonlinear lattices in several ways : the use of a mixed experimental/theoretical approach, incorporation of dissipative effects in the model (although this is done in a rather simplified manner), and a focus on transient energy transport in finite chains (instead of long-time energy spreading in large systems). This study is very interesting and deserves publication after the following points are addressed :

Response: We thank the referee for his/her positive comments and helpful suggestions. Please find below our replies to the specific questions pointed out by the referee.

1- Repeated collision experiments are performed in order to obtain the spatiotemporal distributions of particle velocities. Is it possible that plastic deformations of aluminum beads introduce some bias in the results after too many collisions have occurred?

Response: We thank the referee for this comment. We could not observe any visible dents on the surface of aluminum beads after repeated experiments. We also found that the test data sets do not show significant deviations from each other after repeated tests.

To better understand the plasticity effect, we numerically calculated the von Mises stress of the beads in contact (see Fig. A and Fig. B below). The contact force between particles at the beginning of the chain is around 100 N according to the numerical simulations. However, in the latter part of the chain, the maximum dynamic contact force is mostly less than about 30 N due to the large dispersion and dissipation of the propagating waves. In the case of the contact between the chrome steel and the aluminum beads at 80N (see Fig. A), the maximum Von Mises stress is about 850 MPa, and it appears inside of the sphere (approximately 0.07 mm depth from the contact surface). On the contact surface, the maximum Von Mises stress is about 300 MPa (see the magnified view in Fig. A).

In the case of the contact between the chrome steel striker and the aluminum bead at 100 N, the maximum Von Mises stress in the aluminum bead is around 1300 MPa, and the maximum stress on the bead surface is around 450 MPa (see Fig. B). The yield strength of aluminum depends on the heat treatment and alloying type and can range from 90 MPa to about 690 MPa. However, it is typically around 200 MPa to 400 MPa.

Therefore, we conclude that although plastic deformation may happen inside the aluminum beads, it would have only a minor effect on the surface of the beads.

Von Mises Stress in cross section of sphere contact under 80N of contact force

Fig A. Von Mises stress in Hertz contact between Chrome steel and Aluminum sphere at 80N compression.

Fig B. Von Mises stress in Hertz contact between Chrome steel (striker) and Aluminum sphere at 100N compression.

We also tested for the possibility of plastic effects. First, we note that we did not observe any visible dents on the surface of aluminum beads after repeated experiments. Additionally, the test data sets did not exhibit significant deviations from each other after repeated tests. To examine this possibility further, we also numerically calculated the von Mises stress of the particles in contact (not shown). We found that although plastic deformation may happen inside the aluminum particles, it would have only a minor effect on the particle surfaces.

We added a new Supplementary Note 11 to the supplementary information to briefly discuss this point.

2- What is the influence of impact velocity on the different power-law behaviors ?

Response: We conducted numerical tests using a variety of different impact velocities around 0.45 m/s and observed no significant differences in the exponents. More generally, the presence of both superdiffusive transport and wave localization, which we see in the values of both m_2 and P^{-1} , is robust to changes in the initial condition of the impactor.

We added a comment about this to the Methods section of the paper.

3- At the end of the last section; Methods, it is indicated that the relative error tolerance for ODE45 is set to 10^{-6} m for displacements. Since relative displacements typically lie below one micron in the present physical setting, it may be worthwhile to check numerical computations with a lower error tolerance.

Response: We thank the referee for this comment. From calculations with lower error tolerances, we confirm that the error tolerance that we used (a relative error tolerance of 0.1%) converges to the same results as lower error tolerances. Based on the relative tolerance of 0.1%, the order of magnitude of the errors in displacements is less than $1e-9$ [m], and the order of magnitude of the error in velocities is less than $1e-4$ [m/s] – $1e-6$ [m/s], depending on the specific properties of the waves.

- 4- It could be interesting to mention the following reference dealing with impacts in dissipative granular chains : N.S. Nguyen and B. Brogliato, Multiple impacts in dissipative granular chains, Lect. Notes in Applied and Computational Mechanics 72, Springer, 2014. In this work (see section 5.3), the inverse participation ratio P^{-1} is computed (in the form of an equivalent coefficient $C_{\{KE\}} = \sqrt{N \cdot P^{-1} - 1}$) for different types of granular chains (homogeneous, tapered, diatomic) after impact termination.

Response: We thank the referee for this reference. It is certainly relevant to our study, and we now cite it in the introduction.

Reviewer #2 (Remarks to the Author):

This is an interesting study of wave propagation in disordered granular chains. It probes Anderson localization, and the destruction of Anderson localization due to nonlinearity, whose strength is controlled with the precompression strength. Anderson localization itself is quite a sensitive phenomenon relying on phase coherence, and it is quite surprising that the authors can demonstrate its existence (Fig.3). The crossover from superdiffusive to subdiffusive spreading with increasing precompression is also nice, although the experimental data are actually not (yet) indicating subdiffusion, and the argument of the authors - similar data for few numerical runs, but systematic lowering of the exponent for more averaging over more runs - is not a solid one, reducing that claim to wishful thinking. Therefore the authors will have to change the title accordingly. I must also say that fitting power laws over one decade is not very solid. Probably the large error bars are a consequence of that. To conclude, this paper can be published, but only after the authors change the title, abstract, and the main body of the paper, removing the claim to have measured subdiffusive transport.

Response: We thank the referee for his/her positive comments. Regarding his/her comment on subdiffusive transport, we have amended the main text following the referee's suggestions, and we have also removed any allusions to having experimentally observed subdiffusion in granular crystals from both the title and the abstract. In terms of fit of exponential decay, this is meant just to give a trend for the initial decay, so we adjusted the language in the text (with some phrasing changes in both the main manuscript file and supplementary information) to make it clearer that we are referring to a trend of initial decay.

REVIEWERS' COMMENTS:

Reviewer #3 (Remarks to the Author):

The present consider for the first time (to my knowledge) experimental evidence of Anderson localization effects in mechanical systems. The results presented in the manuscript are of sufficient interest, relevance and impact to warrant publication in Nature Communications.

In my opinion, the authors have successfully replied to the criticism of Referees 1 and 2, and I have only a couple of minor remarks to be taken into account. Once the authors consider these minor changes, the paper should be published in Nature Communications.

The minor remarks are the following ones:

1) In the abstract, it is stated that "disorder and nonlinearity --which are known from decades of studies to individually favor energy localization-- can in some sense cancel each other out". From this sentence, a reader non familiar with the dynamics of nonlinear disordered systems might think that in the paper has been found, for the first time, the nonlinearity can destroy Anderson localization. However, this effect is well known from many previous studies. For this reason, I recommend to soften the claim.

2) At the end of page 7, it is indicated that the equations of motions are integrated using Matlab's ODE45 routine. I think it would be better to indicate that integration has been performed by means of a Dormand-Prince algorithm (which is obviously implemented in Matlab by ODE45 function).

06 January 2018

Dr. Editor,

We thank the referee for his/her positive opinion of our revised manuscript and for his/her additional minor suggestions. We respond to each of his/her points below.

Sincerely,

Mason A. Porter (on behalf of all authors)

Responses to the referee comments:

1) In the abstract, it is stated that "disorder and nonlinearity --which are known from decades of studies to individually favor energy localization-- can in some sense cancel each other out". From this sentence, a reader non familiar with the dynamics of nonlinear disordered systems might think that in the paper has been found, for the first time, the nonlinearity can destroy Anderson localization. However, this effect is well known from many previous studies. For this reason, I recommend to soften the claim. Also, the abstract was too long by about 35 words, so this is also something we should iterate on to make sure we're happy.

Response: We have adjusted the abstract to address this point. Given the editorial comment on the use of quotations in the abstract, we also made some additional minor adjustments. To ensure that the abstract meets the Nature Communications length requirements, we have also made some additional changes to shorten it.

2) At the end of page 7, it is indicated that the equations of motions are integrated using Matlab's ODE45 routine. I think it would be better to indicate that integration has been performed by means of a Dormand-Prince algorithm (which is obviously implemented in Matlab by ODE45 function).

Response: We have added that the ODE45 implements a Dormand-Prince method, as requested by the referee.